# The use of geographic information systems (GIS) in studying mental health service delivery: A scoping review

## Overview Review

digital technologies; efficiency; global mental health; global mental health delivery; health inequalities

**Corresponding author:**
Abhijit Nadkarni;
Email: abhijit.nadkarni@lshtm.ac.uk

Bijayalaxmi Biswal[1], Rakshanda Paimapari[1], Arya Suresh[1], Marimilha Grace Pacheco[1], Luanna Fernandes[1], Yashi Gandhi[1,2], Vikram Patel[3] , Daisy Radha Singla[4] , Anisah Fernandes[1], Richard Velleman[1,5], Chunling Lu[3,6], Chris Grundy[7], Urvita Bhatia[1,2] and Abhijit Nadkarni[1,2]

[1]Addictions and Related Research Group, Sangath, Goa, India; [2]Department of Population Health, London School of Hygiene and Tropical Medicine, London, UK; [3]Department of Global Health and Social Medicine, Harvard Medical School, Boston, MA, USA; [4]Campbell Family Mental Health Research Institute, Centre for Addiction and Mental Health, Toronto, ON, Canada; [5]Department of Psychology, University of Bath, Bath, UK; [6]Division of Global Health Equity, Brigham and Women's Hospital, Boston, MA, USA and [7]Department of Infectious Disease Epidemiology and International Health, London School of Hygiene & Tropical Medicine, London, UK

## Abstract

Geographic information systems (GIS) are computer-based spatial mapping tools widely used in public health to examine service availability and access disparities and healthcare utilization. While GIS has supported evidence-based health planning in various domains, its application in mental healthcare service delivery remains underexplored. Our scoping review aimed to address this gap by exploring the scope and type of GIS usage in studying three dimensions of mental health (MH) service delivery (availability, accessibility and utilization), across all geographical locations, settings and populations. We conducted a scoping review following the Joanna Briggs Institute methodology. We included peer-reviewed English-language studies using GIS to examine service delivery (availability, accessibility or utilization) for any MH condition diagnosed through standardized criteria or validated tools. Seven databases were searched (Medical Literature Analysis and Retrieval System Online [MEDLINE], PsycINFO, Excerpta Medica Database [Embase], Global Health, Cumulative Index to Nursing and Allied Health Literature [CINAHL], Cochrane Central Register of Controlled Trials [CENTRAL] and Web of Science) between January and April 2024. This review included 58 studies predominantly from high-income countries. A wide range of GIS methods were employed across studies, including hotspot analysis, network analysis and spatial analysis. Six studies explored availability, generally through measures like distribution of facilities across a population, and resource availability within 5–10-mile network buffers. Forty-six studies explored the spatial accessibility of MH services and substance-use treatment facilities using GIS. Six studies examined service utilization patterns. Equity emerged as a recurring theme across all three dimensions. GIS has the potential to emerge as a powerful tool in MH research, particularly in mapping disparities, informing service delivery and identifying high-risk zones. Expanding GIS use in trial design, implementation science and policy advocacy could help bridge critical gaps in MH service delivery, ensuring more equitable and data-driven decision-making.

## Impact statement

This scoping review provides a comprehensive synthesis of how geographic information systems (GIS) have been used to study the availability, accessibility and utilization of mental health (MH) services. The findings highlight GIS as a powerful, yet underutilized, tool for identifying gaps in service coverage, visualizing disparities across regions and populations and informing data-driven MH policy and planning. By cataloging a wide range of GIS methods and applications from 58 studies, the review lays critical groundwork for the integration of spatial analysis into global MH research.

The review reveals that GIS has predominantly been applied in high-income settings, with limited application in low- and middle-income countries (LMICs) where treatment gaps are largest. It identifies significant opportunities for expanding GIS use in MH implementation research, trial design and policy advocacy – especially in underserved communities. By uncovering invisible barriers to care through spatial mapping, GIS offers an innovative pathway toward more equitable MH systems.

For policymakers, researchers and practitioners, this review provides both a roadmap and a call to action: to harness the full potential of GIS for strengthening MH services, improving access for marginalized populations and driving evidence-based reforms. The insights from this review can support national and local governments, donors and program implementers in making more informed, targeted and just decisions in mental healthcare delivery.



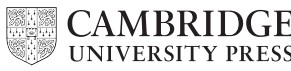

## Introduction

Geographic information systems (GIS) are an innovative computer-based spatial mapping technology that can provide an enhanced understanding of patterns, service needs and environmental interactions related to health problems for improving care (Walsan et al., 2016). These systems are equipped to collect, manage and visualize spatial data, assisting in the analysis and interpretation of geographic information. It can be used to examine, quantify and interpret relationships and features within geographic data (McLafferty, 2003). It has been widely used in the field of public health, especially for understanding the spatial organization of health care, studying healthcare utilization patterns and mapping the availability of healthcare services (McLafferty, 2003; Higgs, 2004, 2009; Graves, 2008). It also has advanced applications in mapping access disparities, disease surveillance, health inequities and emergency responses (Graves, 2008; Higgs, 2009). Through integrated analysis of demographic, environmental and clinical data, GIS has been used to support evidence-based policymaking (Hannum et al., 2025).

Little is known about GIS approaches that have been used in the analysis of mental healthcare service delivery. This has not only precluded a comprehensive understanding of the full potential of GIS in MH research, implementation science, health planning and service delivery but also limited the possibilities of its usage. Leveraging GIS use in exploring mental healthcare service delivery is especially important considering the global focus shifts toward community MH, implementation research and treatment equity gap, which are profoundly shaped by logistical barriers and the practicality of help-seeking (Thornicroft et al., 2016; Kola et al., 2021; Orozco et al., 2022; Adams, 2024; McGinty et al., 2024). Our scoping review aimed to address this gap by exploring the scope and type of GIS usage in studying three dimensions of MH service delivery (availability, accessibility and utilization), across all geographical locations, settings and populations.

According to the World Health Organization (WHO) Health Systems Framework, parameters for monitoring a healthcare service delivery system include (a) availability of services: physical presence of services, encompassing health infrastructure, core health personnel and aspects of service utilization, for example, the proportion of health facilities offering specific services; (b) accessibility: geographic accessibility or spatial accessibility, in terms of commuting time spent and distance traversed to reach healthcare services, for example, the time taken for a service user to drive to the nearest health facility; and (c) utilization: quantification or description of the use of healthcare services by people to study trends, patterns, variations or for other objectives (World Health Organization, 2010, 2014; Penchansky and Thomas, 1981; Carrasquillo, 2013), for example, number of outpatient department visits per 10,000 population per year.

In the current study, we considered these three dimensions of service delivery – namely service availability, accessibility and utilization – because they can also be spatially analyzed, hence providing an opportunity for GIS applications. Drawing from the key stages in the Tanahashi Framework, these three components have been used to identify bottlenecks in service coverage and identify specific barriers to accessing and receiving effective MH care and measuring progress toward universal health coverage in MH (Tanahashi, 1978; De Silva et al., 2014).

An integrative review conducted in 2019 reviewed GIS applications that were used to study mental healthcare services but limited its scope only to services provided for serious mental illnesses and to one dimension of service delivery (accessibility; Smith-East and Neff, 2020). Our scoping review sought to provide a more comprehensive synthesis by mapping how GIS has been applied across three key dimensions of MH service delivery (availability, accessibility and utilization) across diverse contexts, conditions, settings and populations. This broader focus not only enabled a holistic overview of the literature but also revealed methodological and conceptual gaps that must be addressed to strengthen the use of GIS in advancing equitable, evidence-informed MH care.

## Materials and methods

We employed a scoping review methodology, which is designed to map the breadth and nature of the existing literature on a topic (Arksey and O'malley, 2005; Peters et al., 2015). This approach is particularly well-suited to our study because it allows for an exploratory and flexible examination of diverse evidence, identifies key concepts and knowledge gaps and supports the development of future research priorities. The review was conducted in accordance with the Joanna Briggs Institute Methodology for Scoping Reviews (2020) and incorporated the Preferred Reporting Items for Systematic Reviews and Meta-Analysis (PRISMA) extension for Scoping Reviews Checklist (Tricco et al., 2018). The review protocol was published on the Open Science Framework in November 2023 (Registration DOI: 10.17605/OSF.IO/QBPJY).

### Eligibility criteria

Peer-reviewed publications in English were included. There were no restrictions on geographical location, year of publication or target population or on design or methodology. Broadly, the scoping review aimed to explore the evidence based on (1) GIS and its various uses in healthcare service delivery (i.e., accessibility, availability and utilization) and (2) MH conditions. Hence, we included any study that 1) used GIS to analyze geographical data and 2) included any MH condition that was diagnosed using one of the following: (a) Diagnostic and Statistical Manual of Mental Disorders, Fifth Edition (DSM-5), or the International Classification of Diseases (ICD) diagnostic criteria; (b) positive screen on a validated screening tool (e.g., 9-item questionnaire for depression [PHQ-9] and 7-item questionnaire for anxiety [GAD-7]); or (c) clinician diagnosis. We excluded studies that solely used global positioning systems (GPSs) or Google Maps for data collection and did not analyze geographical data.

Only studies focusing on service delivery (utilization, accessibility and availability) of healthcare services were included. *Healthcare services* were defined as any primary, secondary and tertiary health care, as well as community MH services, but not interventions which are not traditionally categorized as health care (e.g., social interventions that improve MH). As mentioned in the introduction, we defined *service availability* as the physical presence of services and encompassed health infrastructure, core health personnel and aspects of service utilization. Related constructs such as service coverage, treatment capacity and equity in service availability were included under the dimension of availability. *Accessibility* was defined primarily as geographic accessibility or spatial accessibility, in terms of commuting time spent and distance traversed to reach healthcare services (Penchansky and Thomas, 1981). We were also interested in exploring the relationship of accessibility with help-seeking and treatment adherence. *Utilization* referred to the quantification or description of the use of healthcare services by

people to study trends, patterns, variations or for other objectives (Carrasquillo, 2013). This dimension also conceptually encompassed disparities in service use, hotspots and cold spots and underlying factors influencing doctor visits or hospital admissions.

Although we limited the definition of "accessibility" primarily to its geographic aspect, we are aware that it is a broader concept determined by other factors that affect one's uptake of health care (Andersen and Newman, 1973). Thus, we used "utilization" as a separate concept to capture studies which might highlight the direct or indirect use of GIS in analyzing any other aspects of MH service delivery, especially non-spatial ones (e.g., acceptability or affordability of services). We also anticipated that exploring the concept "utilization" could help us discover studies that have used GIS to assess inequity or disparities in care and explain variations in healthcare use.

Primary and secondary research papers of any design and methodology (including quantitative and qualitative designs if any) were included if they met the inclusion criteria. Both experimental and quasi-experimental study designs including randomized controlled trials, non-randomized controlled trials and analytical observational studies (prospective and retrospective cohort studies, case–control studies and analytical cross-sectional studies) were considered for inclusion. This review also considered descriptive observational study designs including case reports, case series and descriptive cross-sectional studies for inclusion. We excluded reviews, commentaries and opinion pieces.

### Search strategy

Seven electronic databases were searched: Medical Literature Analysis and Retrieval System Online (MEDLINE), PsycINFO, Excerpta Medica Database (Embase), Global Health, the Cumulative Index to Nursing and Allied Health Literature (CINAHL), Cochrane Central Register of Controlled Trials (CENTRAL) and Web of Science. The search was conducted between January 2024 and April 2024, using search terms under the following concepts: MH conditions (e.g., "depression") and geographical information systems (e.g., "geospatial analysis"). The detailed search strategy for MEDLINE can be found in Supplementary Appendix A, and the search strategies for the other databases were a modification of this strategy based on the requirement of each database. Forward and backward citation chaining of included studies was conducted using Web of Science to find any additional eligible studies not identified through the database search.

### Study selection and data extraction

Search results from all electronic databases were merged and imported into EndNote X9 for the removal of duplicates. After automatic and manual de-duplication, the remaining studies were imported into Covidence, an online software for managing systematic reviews. Papers were also manually screened for duplicates on the Covidence platform. A pair of reviewers (BB and RP) independently screened all titles and abstracts and conducted the full-text screening for eligibility. Conflicts were resolved by a third reviewer (LF).

Forward and backward citation chaining of included studies was conducted at this stage using Web of Science to find any additional eligible studies. A data extraction form was developed a priori on Microsoft (MS) Excel to collect data relevant to the objectives of this review and piloted.

Data were extracted by four pairs of researchers (BB and AS, BB and RP, BB and MGP and BB and AF). Inter-rater reliability among the four pairs of raters for data extraction, as measured by Cohen's Kappa ($\kappa$), was deemed excellent (0.81–0.92). Any disagreements between the reviewers during extraction were resolved through discussion till a consensus was reached.

### Data analysis and quality assessment

To effectively summarize the findings in accordance with the objectives of the review, we conducted a narrative synthesis (Popay et al., 2006). This involved a descriptive analysis of the studies included in the scoping review, using a textual approach to summarize and explain the results of the synthesis (Popay et al., 2006). Studies were categorized under service delivery dimensions, and the processes of GIS usage were described. In line with guidelines for scoping reviews (Peters et al., 2015), we did not conduct quality assessments of the included studies.

### Results

Search results are summarized in Figure 1. Of the 8,142 reports identified, 1945 were duplicates. From the remaining 6,197 papers, we excluded 6,092 that did not meet eligibility criteria at the title and abstract screening stage. In total, 105 full texts were assessed for eligibility. Two studies were excluded at this stage because their objectives did not align with our predefined service delivery dimensions instead focusing on spatial patterns in the prevalence of MH conditions. Based on our eligibility criteria, 47 studies were eligible for inclusion. The forward and backward citation chaining process identified 11 additional eligible studies, leading to a total of N = 58.

### Study characteristics (Table 1)

The 58 included studies were published between 1998 and 2024, with most publications (n = 45 of 58, 77.9%) clustered between 2014 and 2024. The wide majority of studies were conducted in high-income countries (n = 53, 91.4%), with most (n = 41, 70.7%) originating from the United States. Two studies emerged from upper-middle-income countries (South Africa and China) (Bhana and Pillay, 1998; Pang and Lee, 2008) and three from lower-middle-income countries (Nigeria, India and Sri Lanka) (Otun, 2016; Rajapakshe et al., 2019; Roberts et al., 2020). The wide majority (n = 56, 96.6%) employed a cross-sectional design, with two exceptions: one utilizing a prospective chart review (Klimas et al., 2014) and the other using both longitudinal and cross-sectional methods (Cantor et al., 2022). None of the studies reported the use of GIS in MH trials. The data used came from a variety of settings including inpatient, outpatient, emergency departments (EDs), community-based settings and primary care settings. Most (n = 32, 55.2%) examined substance-use disorders (SUDs) (mainly opioid use disorders [OUDs]), while others also focused on serious mental illness (e.g., schizophrenia) and common mental disorders (e.g., depression and anxiety). The significant number of papers that focused on SUDs mainly examined OUDs and associated treatment, including medication-assisted treatment (MAT) options (methadone, buprenorphine and naloxone distribution), opioid treatment programs in various settings (clinics and pharmacies) and outpatient treatment for OUD.

Accessibility was the most frequently examined service delivery dimension, with 46 out of 58 studies focusing on this aspect,

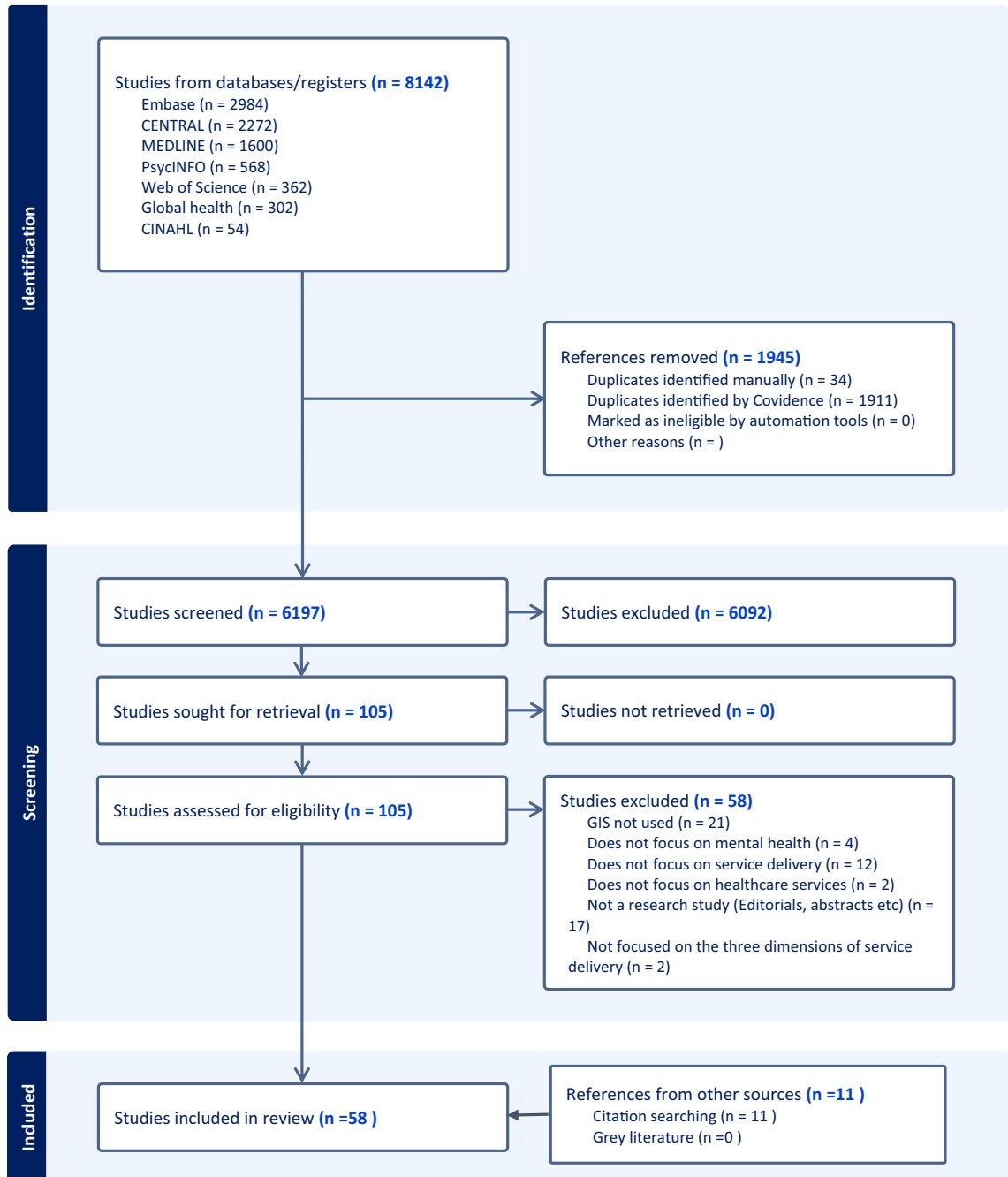

**Figure 1.** PRISMA flow diagram of included and excluded studies.

followed by availability (n = 6) and utilization (n = 6). Types of GIS analysis utilized included variations of spatial analysis (descriptive spatial analysis, spatial regression models and spatiotemporal analysis), hotspot analysis, network analysis, 2-step floating catchment area (2SFCA) method, drive-time comparisons and cluster analysis. Sources of data included provider/specialist directories, Substance Abuse and Mental Health Services Administration (SAMHSA) database, community surveys, inpatient databases, ED databases and census data.

Figure 2 illustrates the conceptual framework used to organise the review findings. The framework builds on the WHO's Service Coverage Framework and the Tanahashi model of health service delivery, adapted to mental health and GIS contexts.

The following section summaries the results into the three key dimensions of accessibility, availability and utilization. In a fourth theme ("Impact"), we report studies that examined how a service delivery dimension impacted other treatment outcomes.

*Availability*

Six studies (Pang and Lee, 2008; Perron et al., 2010; Goedel et al., 2020; Sutarsa et al., 2021; Nolen et al., 2022; Oluyomi et al., 2023) explored availability, generally through measures such as the distribution of facilities across a population and resource availability within 5–10-mile network buffers. Analyses commonly used were hotspot analysis or cluster analysis. Service availability could be further organized as "coverage" and "equity."

**Table 1.** Summary characteristics of included studies

| Author, year | Country | Mental health condition examined | Service examined | Service delivery dimension | Methods | Results |
|---|---|---|---|---|---|---|
| Abell-Hart et al. (2022) | United States | Opioid use disorders and overdose | Naloxone pharmacies and buprenorphine prescribers | Accessibility | Measured distance from the residence to nearest naloxone pharmacy and buprenorphine prescriber | Study identified several geographic hotspots with poor access to naloxone and buprenorphine |
| Alibrahim et al. (2022) | United States | Opioid use disorder | Methadone services and counseling services | Accessibility | Measured estimated driving time for service user from services | Average EDT was 11.32 min, and higher accessibility was observed for counseling services (15.68 min) than methadone services |
| Amiri et al. (2018) | United States | Opioid use disorders | Outpatient treatment for methadone treatment program | Accessibility | Measured travel distance for service user from services | Increased distance (>10 miles) was associated with a higher number of missed doses, indicating lower treatment adherence |
| Amiri et al. (2020) | United States | Opioid use disorders | Methadone treatment in OTPs | Accessibility | Measured travel distance for service user from OTPs | Greater OTP distance linked with missed doses |
| Amiri et al. (2021) | United States | Opioid use disorders | Opioid treatment programs and office-based buprenorphine treatment | Accessibility | Used a 2-step catchment area technique with a distance decay function to study accessibility | Lower access scores were found in more deprived and less urbanized areas (micropolitan and small towns had lower access scores to OTPs) |
| Amram et al. (2019) | Canada | Opioid use disorders | Methadone maintenance treatment (MMT) clinics and federally qualified health centers | Accessibility | Mapping was used to examine areas showing OD clusters | This study found that higher availability of methadone clinics was associated with decreased odds of living within OD clusters |
| Anwar et al. (2022) | United States | Opioid use disorders | Methadone maintenance treatment (MMT) clinics and federally qualified health centers | Accessibility | Assessed % of population within 15- and 30-min drive times from facilities | FQHCs provided greater population coverage within 15–30-min drive times compared to methadone clinics. Methadone clinics had low coverage in high opioid overdose death rate counties |
| Bensley et al. (2021) | United States | Alcohol use disorder (AUD) | Outpatient alcohol treatment | Accessibility | Measured distance and travel time to nearest treatment | Lower treatment density in border cities was associated with a lower likelihood of considering getting help |
| Bhana and Pillay (1998) | South Africa | Mental health conditions in general | Outpatient treatment at mental health clinics and hospitals | Accessibility | Conducted catchment area mapping by defining each catchment as an area within a 5-km radius from a facility | Significant variation found in accessibility across different regions and demographic groups. Urban areas better served than rural areas |
| Burrell et al. (2017) | United States | Overdose deaths related to opioids and other substances | Pharmacy-based intervention, including distribution of naloxone | Accessibility | Compared overdose rates in ZCTAs with naloxone pharmacies to those with non-naloxone pharmacies | Overdose death rates were higher in ZCTAs with naloxone-carrying pharmacies |
| Cantor et al. (2022) | United States | Substance use disorders, opioid use disorder | SUD treatment facilities, forms of payment accepted by these facilities | Accessibility | Measured accessibility as facilities being within 15-, 30- and 60-min driving time for service user and accepting their form of payment | Medicare beneficiaries have less geographic accessibility to SUD treatment facilities compared to users with other forms of payments |
| Charlesworth et al. (2024) | United States | Mental health conditions in general | Access to mental health prescribers and non-prescribers | Accessibility | Calculated 30- and 60-min drive times and E2SFCA access score | Urban areas had higher accessibility and availability, compared to rural and frontier areas |
| Taylor et al. (2017) | United States | Substance use disorder, mental health disorders, opioid use disorder | Emergency department services for overdose and related care | Accessibility | Calculated percentage of all the visits at ED that were opioid-related and formed spatial clusters of ED visits | Identified hotspots or high density in specific tracts indicating higher opioid-related healthcare needs |

*Cambridge Prisms: Global Mental Health*

**Table 1.** (*Continued*)

| Author, year | Country | Mental health condition examined | Service examined | Service delivery dimension | Methods | Results |
|---|---|---|---|---|---|---|
| Dworkis et al. (2018) | United States | Opioid-related mental health issues, opioid overdoses | Emergency medical service runs for opioid overdoses | Accessibility | Used geospatial analysis to examine for clustering in general and to identify specific clusters amenable to publicly deployed naloxone sites | Identified three main clusters where 40% of overdoses occurred within 200 m of cluster centers |
| Goedel et al. (2020) | United States | Opioid use disorders | Medications for opioid use disorders (specifically methadone and buprenorphine) | Availability | Measured rates of methadone and buprenorphine use among individuals with opioid use disorders. Also calculated the number of facilities per 1,00,000 population | Greater methadone access found in counties with high African American/Hispanic segregation; greater buprenorphine access in predominantly white segregated areas |
| Green et al. (2013) | United Kingdom | Common mental disorders | Psychological therapies | Accessibility | Mapped number of referrals to psychological therapy service, distance to service providers and areas of deprivation | Quality improvement initiatives led to a significant increase in referrals, particularly deprivation areas, indicating improved utilization of services |
| Guerrero et al. (2011) | United States | Substance use disorder (SUD) | Spanish-language SUD treatment facilities | Accessibility | Measured road distance to nearest Spanish-language SUD service | Key hotspots found >2.7 miles from services |
| Guerrero et al. (2013) | United States | Substance use disorder (SUD) | Outpatient SUD treatment in Spanish | Accessibility | Measured street-level distance from treatment | Spanish-language facilities averaged 2.74 miles from high-density Latino areas |
| Han and Stone (2007) | United States | Depression and substance use disorders | Psycho-social services and related social services | Accessibility | Calculated travel time and travel distance to explore accessibility | Youth-reported negative neighborhood quality weakly predicted decreased likelihood of psycho-social service receipt |
| Holmes et al. (2022) | United States | Opioid-related overdose incidents | Naloxone administration during opioid overdose incidents | Utilization | Calculated frequency of naloxone administration by country and population density | Higher administration in urban than rural counties; lower rates in predominantly White, middle-aged, rural populations |
| Iloglu et al. (2021) | United States | Opioid use disorder | Methadone treatment for opioid use disorder | Accessibility | Assessed drive time of 15 min to the methadone treatment facility | Study found that one-third of opioid use treatment needs in Ohio were not covered by existing OTPs and the portion of need covered decreased with increasing rural zip code classification |
| Joudrey et al. (2020) | United States | Opioid use disorder | Methadone dispensing services for opioid use disorders and pharmacy-based methadone dispensing locations | Accessibility | Assessed minimum drive time in minutes from the census tract mean center of population to the nearest methadone dispensing facility | Rural census tracts had significantly longer drive times to OTPs compared to urban tracts. Pharmacy-based dispensing could significantly reduce drive times, especially in rural areas |
| Kao et al. (2014) | United States | Drug use (long-term heroin use consequences) | Outpatient substance-use and treatment facilities and methadone maintenance treatment programs | Accessibility | Examined the distance from residence to the closest drug treatment facility (in minutes) and number of facilities within a 10-min driving distance from residence | Increased spatial accessibility was associated with decreased worries about injecting in the future, particularly among current users. The results also suggest that individuals reporting a very high chance of injecting in the future tended to live closer to a facility, as well as in areas with a greater number of facilities |
| Katayama et al. (2023) | United States | Mental health conditions in general | Psychiatric services, access to psychiatrists | Accessibility | Calculated a 30-min drive-time radius around each psychiatrist location to estimate the population served | Only 6% of counties had "convenient" access, meaning the entire population resided within a 30-min drive of a psychiatrist. Rural areas had significantly lower access to psychiatrists compared to urban areas |

(*Continued*)

| Author, year | Country | Mental health condition examined | Service examined | Service delivery dimension | Methods | Results |
|---|---|---|---|---|---|---|
| Kleinman (2020) | United States | Opioid use disorders | Opioid treatment programs (OTPs) versus pharmacies | Accessibility | Measured drive time in minutes from treatment | Driving times to opioid treatment programs and pharmacies in the United States. Mean time to OTPs was 20.4 min versus 4.5 min to pharmacies |
| Klimas et al. (2014) | Ireland | Opioid overdose | Prehospital emergency medical services (EMS) | Accessibility | Measured accessibility through the proximity of ambulance services and addiction services to overdose locations | The study found that overdoses were concentrated in specific areas, particularly in the city center. Overdoses were more likely to occur in areas with higher levels of deprivation and closer to addiction services |
| Koizumi et al. (2009) | United States | Serious mental illness | Community mental health programs | Accessibility | Measured accessibility using 2-step floating catchment area (2SFCA) score | Significant disparities in accessibility across urban and suburban DAs |
| Langabeer et al. (2020) | United States | Opioid use disorder | Buprenorphine-waivered providers | Accessibility | Used geospatial distance buffering analyses to estimate percent of population who are within reasonable (10, 30 and 50 miles) driving distances from a buprenorphine provider | Sparse access found in rural and frontier zones, revealing significant provider gaps in high-need areas |
| Law and Perlman (2018) | Canada | Mental health conditions in general | Doctor visits and hospital admissions | Utilization | Measured utilization by number of doctor visits and hospital admissions | Identified hotspots and cold spots, areas with high hospital admission rates and low doctor visit rates and common risk factors influencing both doctor visits and hospital admissions |
| López-Lara et al. (2012) | Spain | Mental health conditions in general | Mental health services in general | Accessibility | Measured temporal accessibility, that is, travel time to the nearest mental health facility | The study identified areas with limited access to mental health services, particularly in rural regions. It proposed optimal locations for new facilities to improve accessibility for a larger population |
| Metraux et al. (2012) | United States | Severe mental illness (SMI) | Community resources including mental health services, supermarkets and public transport | Accessibility | Measured mean Euclidean distance to each resource type (e.g., mental health service and grocery store) | This study found that a large group of Medicaid recipients diagnosed with SMI had better outcomes, when compared to a representative distribution of Philadelphia locations, on measures of geographic proximity and availability for resources considered to be important to people diagnosed with SMI |
| Mitchell et al. (2022) | United States | Opioid use disorder | Accessibility to opioid treatment programs | Accessibility | Employed a gravity-based variant of the enhanced two-step floating catchment area (E2SFCA) model to measure the accessibility of opioid treatment services. It included distance decay function, provider supply and population density | Rural areas had lower accessibility to services compared to urban areas due to factors such as lower provider density, longer travel distances and limited transportation options |
| Ngamini Ngui and Vanasse (2012) | United States | Mental health conditions in general | Public mental health facilities | Accessibility | Used the 2SFCA method to compute the ratio of suppliers to residents within a service area centered at a supplier's location and sums up the ratios for residents living in areas where different provider's services overlap | Showed that accessibility scores vary greatly from one DA to another |
| Nolen et al. (2022) | United States | Opioid-related overdose death | Overdose antidote naloxone | Availability | Used geospatial methods to calculate naloxone coverage ratios for each municipality in two states of the United States | Found no municipal-level racial/ethnic inequities in naloxone distribution in Rhode Island and Massachusetts, United States |

**Table 1.** (*Continued*)

| Author, year | Country | Mental health condition examined | Service examined | Service delivery dimension | Methods | Results |
|---|---|---|---|---|---|---|
| Oluyomi et al. (2023) | United States | Obsessive–compulsive disorder | Cognitive-behavioral therapy (CBT) for obsessive–compulsive disorder (OCD) | Availability | Examined the geographic distribution of OCD–CBT specialty providers across the state of Texas | Specialist providers are almost exclusively located in highly urbanized parts of the state. Characteristics of areas located furthest away include persons identifying as Hispanic; non-English speakers, households with income below poverty and persons with no health insurance |
| Otun (2016) | Nigeria | Any mental health condition | Mental health services in general | Accessibility | Used coordinates of the location of the mental health facilities and settlements to examine accessibility to mental health care | 74.85% of the settlements are more than 10 km from the nearest MHC |
| Pang and Lee (2008) | China | Heroin addiction | Methadone treatment program (MTP) | Availability | Used a simplified methodological framework to measure the geographic coverage of methadone clinics | The average geographic coverage in Hong Kong is 44.6%, with the figure varying from 0% to 96% by district |
| Perlman et al. (2018) | Canada | Cognitive disorders, mainly delirium, dementia and amnesia | Inpatient psychiatry admission for any condition | Accessibility | Examined accessibility of general hospitals with psychiatric beds and psychiatric hospitals by calculating distance for the service user | Accessibility to hospitals was marginally insignificant at the 95% credible interval in the final model. Risk of admission was positively associated with residential instability and the overall hospitalization rate, but not distance to the closest general or psychiatric hospital |
| Perron et al. (2010) | United States | SUDs | Outpatient SUD treatment programs | Availability | Examined geographic accessibility to receive outpatient SUD treatment | There may be an urban bias in SUD treatment programs, which ignores actual living patterns and thus reduces accessibility for certain population clusters |
| Pustz et al. (2022) | United States | Opioid use disorders | Opioid treatment programs and buprenorphine providers | Accessibility | Used drive-time maps and an accessibility index to describe access to substance-use treatment and harm reduction services | Accessibility to these clinicians was limited to urban centers. Most individuals lived further than a four-hour round-trip drive to the nearest methadone treatment program |
| Rajapakshe et al. (2019) | Sri Lanka | Mental health conditions in general | Mental health services in general | Accessibility | Developed an accessibility map and superimposed it on the elderly population density map to examine the accessibility coverage | Certain denser areas of elderly populations in the western parts of the district were not covered by the centers. The traveling time with high congestion of traffic emerged as an identified issue |
| Rhew et al. (2023) | United States | Dementia | Inpatient hospitalization and emergency department healthcare utilization | Utilization | Used existing datasets to profile hospital admission rates and ED visit rates stratified by rurality and regions | Minnesota rural areas showed a 17.6% lower age-adjusted rate (AAR) of dementia mortality than urban areas |
| Roberts et al. (2020) | India | Depression | Mental health services | Accessibility | Examined accessibility using travel distance from households to the nearest public depression treatment provider | Found no association between travel distance and the probability of seeking treatment for depression. Those living in the immediate vicinity of public depression treatment providers were just as unlikely to seek treatment as those living >20 km away by road |
| Schneider et al. (2020) | United States | Opioid use disorder | Emergency and rehabilitation services related to opioid overdose | Accessibility | Assessed maximum distance to an emergency department from each town and summed with overdose scores to obtain overall risk score for each town | Identified towns with high overall risk score. Results also show that distance to both emergency and rehabilitation resources affects outcomes in patients with opioid use disorders |

| Author, year | Country | Mental health condition examined | Service examined | Service delivery dimension | Methods | Results |
|---|---|---|---|---|---|---|
| Schwarz et al. (2022) | Germany | Intensive home treatment | Inpatient hospital treatment and inpatient equivalent home treatment (IEHT) | Utilization | Conducted spatial analyses to study the extent to which the location of the service user's home within the catchment area, as well as the distance between the home and the clinic, influences the utilization of two treatment models | The mean travel times and distances to the place of residence only differed minimally between the two groups. The places of residence of substance users treated with IEHT were located in greater proximity to each other than those treated in inpatient setting |
| Simmons (2019) | United States | Serious mental illness | Publicly funded mental health services | Accessibility | Conducted an optimized hotspot analysis to determine which regions were the most underserved in terms of serious mental illness | The distribution of high burden of serious mental illness areas correlated with neighborhood poverty |
| Sutarsa et al. (2021) | Australia | Mental health conditions in general | Mental health nurses | Availability | Measured the availability of mental health nurses using total full-time equivalent (FTE) rates per 1,00,000 population and proportion of local government areas (LGAs) with zero total FTE rates | A significant proportion of LGAs, particularly in remote and very remote areas, had zero FTE mental health nurses. The average FTE rate for mental health nurses was lower in remote and very remote areas compared to major cities |
| Thurston and Freisthler (2020) | United States | Opioid use disorder | Emergency medical services (EMS) response to opioid overdoses, specifically the administration of naloxone | Utilization | Assessed geographic distribution of EMS stations and response times, availability of naloxone within EMS vehicles and at other locations, policies and protocols regarding naloxone administration by EMS personnel | Naloxone events were clustered in specific geographic regions of rural Ohio, near major highways and interstates |
| Topmiller et al. (2018) | United States | Opioid use disorder | Medication-assisted treatment for opioid use disorder | Accessibility | Focused on identifying areas with limited access to MAT providers, as measured by the number of DEA-waivered practitioners per 1,00,000 population | Identified 29 opioid dependence priority areas, 11 unmet treatments need priority areas and seven low MAT capacity priority areas, located across the United States |
| Townley et al. (2018) | United States | Schizophrenia spectrum of major affective disorder | Outpatient treatment | Accessibility | Examined the relationship between community participation and resource accessibility (i.e., proximity) and availability (i.e., concentration) among individuals utilizing community mental health services throughout the United States | Findings suggested small but significant associations between community participation and the accessibility and availability of resources needed for participation |
| Upadhyay et al. (2019) | United States | Pediatric depression | Depression treatment (antidepressants and psychotherapy) | Accessibility | Measured travel distance from residence and the provider density within a 5-mile radius of each patient to explore how both these factors were associated with treatment engagement | Results of multivariate logistic regression analysis demonstrated that travel distance to provider was negatively associated with the treatment engagement of Hispanics, while a higher mental health specialist density was positively associated with the treatment engagement of Blacks. Among those who have engaged in the treatment, travel distance was associated with a lower likelihood of treatment completion in all racial/ethnic groups |
| Wani et al. (2019) | United States | Substance use disorders (SUDs) | Substance-use treatment facilities and ED visits | Accessibility | Measured spatial distribution and density of EDs and treatment centers across counties in NY | Inequities were found, with urban areas showing higher availability of EDs but also a higher frequency of SUD-related visits |
| Wei and Chan (2021) | Taiwan | Opioid use disorder | Opioid agonist therapy (OAT) | Accessibility | Investigated the association between distance to the treatment site and choice of OAT | Multivariate logistic regression was used to assess the correlation between individual drug selection and distance of residence. Patients living closer to |

**Table 1.** (*Continued*)

| Author, year | Country | Mental health condition examined | Service examined | Service delivery dimension | Methods | Results |
|---|---|---|---|---|---|---|
| | | | | | | the treatment center were more likely to choose methadone as treatment, while patients living farther away were more likely to choose sublingual buprenorphine |
| Winckler et al. (2023) | United States | Acute pediatric mental health (MH) | Acute pediatric mental health (MH) interventions or services | Utilization | Measured mental health (MH) utilization by calculating the number of MH visits per 1,000 children in each census tract | ED and hospital utilization for pediatric MH concerns varied significantly by neighborhood and demographics. Divergent social factors mapped onto these locations and were related to MH utilization |
| Wong et al. (2010) | United States | Mental health conditions in general | Primary health care for mental health, including availability of physicians and clinics | Accessibility | Measured travel time and distance to nearest primary care facilities for mental health | Accessibility varied significantly across neighborhoods, with lower accessibility in lower-income areas |
| Wootten et al. (2024) | Canada | Psychotic disorders in association with cannabis use | Health service use for psychosis outpatient visits, emergency department (ED) visits and hospitalizations | Accessibility | Calculated standard walking distance and driving distance to cannabis retail outlets and examined relationship of accessibility to outlets with service use | Living in proximity to cannabis retail outlets was associated with higher rates of outpatient visits, ED visits and hospitalizations for psychotic disorders |
| Yen and Lin (2019) | Taiwan | Dementia | Dementia care providers | Accessibility | Measured "tolerance limited distance (TLD)," that is, the maximum distance a user is willing to travel to access dementia care services | Identified areas with high TLDs. Areas with lower TLDs were considered to have better accessibility, as users were willing to travel shorter distances |
| Zulian et al. (2011) | Italy | Mental health conditions in general | Acute inpatient wards, community mental health centers (CMHCs) and outpatient clinics | Accessibility | Measured geographic proximity of patients to services using distances calculated along the road network | Facilities were unevenly distributed, with rural areas underserved. A distance decay effect showed decreased service use with increased distance: a 1.5% decrease for inpatient wards, 2.0% for CMHCs and 2.1% for outpatient clinics per service area increase in distance. |

**Figure 2.** The conceptual framework used to organize the review findings. The framework builds on the WHO's Service Coverage Framework and the Tanahashi model of health service delivery, adapted to mental health and GIS contexts.

**Service coverage.** Out of the six, three studies (Pang and Lee, 2008; Sutarsa et al., 2021; Nolen et al., 2022) explored availability in terms of treatment/service coverage. Nolen et al. (2022) used naloxone coverage ratios (the number of naloxone kits distributed through community-based programs to the number of opioid-related overdose deaths among its residents) to determine if US municipalities with high percentages of racial minorities have equitable access to the overdose antidote naloxone (Nolen et al., 2022). Pang and Lee (2008) used district-based geographic coverage to evaluate the methadone treatment program (MTP) in Hong Kong (Pang and Lee, 2008). Sutarsa et al. (2021) investigated the spatial distribution of MH nurses across Australian local government areas by measuring the number of full-time equivalent MH nurses per 1,00,000 people, revealing significant regional disparities (Sutarsa et al., 2021).

**Equity in service availability.** The remaining three studies (Perron et al., 2010; Goedel et al., 2020; Oluyomi et al., 2023) focused on equity in service availability. Two studies (Perron et al., 2010; Goedel et al., 2020) attempted to evaluate treatment capacities of particular regions by identifying the distribution of healthcare facilities, determining population covered by service catchment areas and calculating the total number of resources within 5–10-mile Euclidean buffers from patients' addresses (i.e., straight-line distances from patients' addresses). One study examined the geographic distribution of OCD CBT speciality

providers across the state of Texas, with particular attention to the relationship with neighborhood socioeconomic disadvantage, insurance status and rural versus urban status (Oluyomi et al., 2023).

*Accessibility*
Forty-six studies aimed to explore the spatial accessibility of MH services and substance-use treatment facilities using GIS. Accessibility was most commonly defined as the ease with which individuals can reach and utilize MH services. Temporal accessibility, measured by travel time to the nearest MH facility, and spatial accessibility, measured by distance to nearest facility, were generally used measures to assess accessibility, with a small number of studies also using parameters like population within a convenient distance of services (5–10 miles from a facility or within a 30-minute drive from healthcare services). Only one of the studies used the cost of travel as a metric (Han and Stone, 2007). Studies relied on usual data sources (census data, the SAMHSA database, community surveys, etc.) occasionally using them alongside databases linked to law or justice departments like the Drug Enforcement Administration (DEA).

**Equity in service accessibility.** A substantial number of these papers (n = 16) focused on studying equity of services (Bhana and Pillay, 1998; Koizumi et al., 2009; Perron et al., 2010; Guerrero et al., 2011; López-Lara et al., 2012; Guerrero et al., 2013; Amiri

et al., 2018; Rajapakshe et al., 2019; Simmons, 2019; Upadhyay et al., 2019; Wani et al., 2019; Joudrey et al., 2020; Langabeer et al., 2020; Pustz et al., 2022; Katayama et al., 2023; Charlesworth et al., 2024). Twelve focused on the rural–urban divide of mental healthcare services, using spatial analysis to visually map areas with limited access to MH services with help of rural and urban census tracts (Bhana and Pillay, 1998; Koizumi et al., 2009; Perron et al., 2010; López-Lara et al., 2012; Amiri et al., 2018; Upadhyay et al., 2019; Wani et al., 2019; Joudrey et al., 2020; Langabeer et al., 2020; Pustz et al., 2022; Katayama et al., 2023; Charlesworth et al., 2024) and generally concluding that rural areas were underserved compared to urban areas. In addition to the usual 30-minute or 60-minute drive times, some studies also used other methods of calculating access like enhanced two-step floating catchment area (E2SFCA) method access score (Charlesworth et al., 2024), 2SFCA technique with a distance decay function (Amiri et al., 2018), geospatial distance buffering (Langabeer et al., 2020) and network analysis (Roberts et al., 2020). After visually mapping accessibility, two studies used spatial regression techniques to explore associations with socio-demographic factors that further determined healthcare access (Perron et al., 2010; Amiri et al., 2018). The other four studies studied equity of services by focusing on access for vulnerable populations (elderly, ethnic minorities, socio-economically weak groups) (Guerrero et al., 2011; Guerrero et al., 2013; Rajapakshe et al., 2019; Simmons, 2019). One study used network analysis methods to map dementia care service points geographically with relation to elderly population density (Rajapakshe et al., 2019). Simmons (2019) conducted an optimized hotspot analysis to determine which regions were the most underserved in terms of serious mental illness burden and correlated it to neighborhood poverty (Simmons, 2019). Two studies assessed the distance between Latino-populated census tracts and general MH treatment facilities (Guerrero et al., 2011; Guerrero et al., 2013).

**Opioid dependence and accessibility of treatment.** A number of papers (n = 12) used different methods to explore the same objective: identifying high-risk zones for opioid dependence in the United States and exploring accessibility of emergency services and inpatient and outpatient treatment for the same. Eight studies mapped overdose incidents and compared them to the location of treatment services (ambulance services and methadone/naloxone facilities), highlighting areas of deprivation and concluding that having a treatment facility within 15- and 30-minute drive time from hotspots of overdose deaths was associated with lower risks of overdoses (Kao et al., 2014; Klimas et al., 2014; Burrell et al., 2017; Taylor et al., 2017; Dworkis et al., 2018; Amram et al., 2019; Iloglu et al., 2021; Anwar et al., 2022). One study tried to obtain an overall risk score by summing distance scores and overdose scores for each town in a state to create a map which approximated the need for additional emergency resources by town (Schneider et al., 2020). After identifying high-risk areas, they further examined how the inaccessibility of resources affects outcomes in patients with OUDs. One study mapped opioid dependence priority areas and areas with low numbers of DEA-waivered practitioners to identify unmet treatment need priority areas and low MAT capacity priority areas (Topmiller et al., 2018). Kleinman (2020) used population-weighted mean travel time from census tracts to nearest opioid treatment programs and pharmacies, comparing two models of methadone dispensing and demonstrating that pharmacies were more accessible for this purpose than opioid treatment programs (Kleinman, 2020). Abell-Hart et al. (2022) identified several

hotspots where patients lived far from naloxone/buprenorphine providers (Abell-Hart et al., 2022).

**Accessibility and help-seeking.** Two studies examined accessibility and its association with demand for care or help-seeking. Bensley et al. (2021) explored the distance and travel time to nearest treatment services (using network analysis) to show that lower service density was associated with a lower likelihood of considering getting help (Bensley et al., 2021). Conversely, Roberts et al. (2020) found no association between travel distance and the probability of seeking treatment for depression (Roberts et al., 2020).

**Accessibility and treatment adherence.** Three studies explored the relationship between treatment accessibility and adherence. Two studies concluded that increased distance (>10 miles) was associated with a higher number of missed doses or lower treatment adherence (Amiri et al., 2018; Amiri et al., 2020), while another used multivariate logistic regression analysis to demonstrate the relationship between travel distance and treatment completion for minority groups (Upadhyay et al., 2019).

*Utilization*

Six studies examined service utilization patterns (Perlman et al., 2018; Thurston and Freisthler, 2020; Holmes et al., 2022; Schwarz et al., 2022; Rhew et al., 2023; Winckler et al., 2023) with two studies focusing on equity of services or disparities (Holmes et al., 2022; Rhew et al., 2023).

**Equity in service utilization.** Rhew et al. (2023) studied rural–urban differences in healthcare utilization for older adults with dementia across the state by exploring hospital admission rates and ED visit rates related to dementia, stratified by rurality and regions (Rhew et al., 2023). Holmes et al. (2022) explored disparities in opioid overdose survival and naloxone administration across different counties in Pennsylvania (Holmes et al., 2022).

**Patterns of service use.** Thurston and Freisthler (2020) examined the frequency and geographic distribution of EMS calls resulting in naloxone administration (Thurston and Freisthler, 2020). Schwarz et al. (2022) studied the extent to which the location of the service user's home within the catchment area, as well as the distance between the home and the clinic, influences the utilization of two treatment models (inpatient treatment compared to IEHT) (Schwarz et al., 2022). Winckler et al. (2023) measured the rate of MH visits per 1,000 children in specific geographic regions (census tracts) to assess the extent to which MH services were being accessed and used by the target population with the aim of identification of high utilization for the pediatric population (Winckler et al., 2023). Perlman et al. (2018) examined the geographic variation in MH service utilization in Toronto at the neighborhood level identifying hotspots and cold spots, spatial patterns and underlying factors measured by doctor visits and hospital admissions (Perlman et al., 2018).

*Impact*

Seven studies examined how a service delivery dimension (availability, accessibility or utilization) impacted other outcomes (Kleinman, 2020; Thurston and Freisthler, 2020; Wei and Chan, 2021; Alibrahim et al., 2022; Cantor et al., 2022; Schwarz et al., 2022; Charlesworth et al., 2024).

**Program or policy evaluation.** Thurston and Freisthler (2020) examined the frequency and geographic distribution of EMS calls resulting in naloxone administration and identified clusters of naloxone events (Thurston and Freisthler, 2020). They eventually concluded that spatial clusters crossed administrative boundaries (i.e., county lines) suggesting that opioid misuse was less responsive to county-level policies. Cantor et al. (2022) assessed the proportion of individuals who had a SUD treatment facility within a 15-minute drive that accepted their specific form of payment – Medicaid, private insurance or cash (Cantor et al., 2022). The study found that Medicaid beneficiaries faced lower geographic accessibility to SUD treatment services, primarily because fewer facilities accepted Medicaid compared to other payment types.

**Impact on treatment choices.** Five studies showed how accessibility influenced treatment choices (Kleinman, 2020; Wei and Chan, 2021; Alibrahim et al., 2022; Schwarz et al., 2022; Charlesworth et al., 2024). One study compared driving time from zip codes of patients to treatment facilities to show that higher accessibility was observed for counseling services than methadone services (Alibrahim et al., 2022). Wei and Chan (2021) compared the distance between the patients' residence and treatment centers to discover that patients living closer to the treatment center were more likely to choose methadone as treatment, while patients living farther away were more likely to choose sublingual buprenorphine tablets (Wei and Chan, 2021). Another study investigated the extent to which the location of the service user's home within the catchment area, as well as the distance between the home and the clinic, influences the utilization of inpatient treatment compared to inpatient equivalent home treatment (IEHT) (Schwarz et al., 2022). Kleinman (2020) used population-weighted mean travel time from census tracts to nearest opioid treatment programs and pharmacies, comparing two models of methadone dispensing and demonstrating that pharmacies were more accessible for this purpose than opioid treatment programs (Kleinman, 2020). Charlesworth et al. (2024) examined access to MH prescribers and non-prescribers in rural areas and found that mental healthcare delivery in rural settings often relied on non-prescribers, owing to limited access to Medicaid-participating prescribers (Charlesworth et al., 2024).

## Discussion

To our knowledge, this is the first scoping review to comprehensively synthesize how GISs have been applied across three core dimensions of MH service delivery spanning diverse populations, settings and geographical regions. Our review builds upon previous literature by moving beyond a narrow focus on serious mental illness and accessibility to encompass a broader spectrum of MH conditions and service delivery dimensions. The findings not only demonstrate a growing literature in GIS applications of MH service delivery but also point to a highly uneven distribution of research (both thematically and geographically) with a concentration of studies in high-income countries and a predominant focus on spatial accessibility. This review has identified several underexplored areas in the application of GIS that have the potential to advance MH service planning and delivery globally, including its use in designing and monitoring clinical trials, supporting implementation research and informing advocacy strategies.

## Current scope and patterns of use across studies

About one-third of eligible studies across all three themes had primary objectives related to resource management and planning, focusing on identifying high-risk zones or priority areas for opioid dependence, hotspots of overdose deaths or unmet treatment needs, mapping them against areas where treatment services or providers are located. Treatment or service coverage, another metric of importance to resource planning, was explored by conducting spatial analyses of services delivered in comparison with the target population. In addition to quantifying service gaps, studies focused on this theme also suggested potential interventions, such as expanding treatment infrastructure or modifying service delivery models to enhance access.

Another emerging focus that is consequential for resource allocation was studying equity of services (n = 25), which was explored by looking at disparities in service delivery for marginalized populations and rural/urban areas. Furthermore, these studies explored structural inequities by assessing associations between spatial healthcare access and socioeconomic indicators, race/ethnicity and insurance status, highlighting systemic barriers and advocating for equity-driven policy reforms.

Some studies used GIS for program evaluation or policy impact assessment, like comparing two different models of methadone maintenance programs (Iloglu et al., 2021) and the restrictive payment model of Medicaid (Charlesworth et al., 2024), often suggesting equity-informed interventions and changes in policy (Kleinman, 2020; Cantor et al., 2022).

The strength of existing databases and electronic health records emerged as a major determinant of GIS usage, which could possibly explain why only three studies were conducted in low- and middle-income countries (LMICs) (Otun, 2016; Rajapakshe et al., 2019; Roberts et al., 2020). GIS applications in MH research relied heavily on existing databases, including census data, provider directories, community surveys and law enforcement databases. Some of these databases also helped facilitate real-time tracking of healthcare trends, enabling analysis without the need for additional primary data collection. The integration of multiple data sources, such as the SAMHSA database, DEA reports and ED records, allowed for a more comprehensive analysis of MH service distribution (Topmiller et al., 2018; Kleinman, 2020; Iloglu et al., 2021; Abell-Hart et al., 2022; Charlesworth et al., 2024). Interdisciplinary approaches, such as combining healthcare data with law enforcement statistics, helped studies enhance the scope of their analysis and provide a multidimensional perspective on MH service accessibility and availability (Adelfio et al., 2019).

The predominance of opioid-related GIS studies conducted in the United States could be explained by the presence of strong surveillance infrastructure and the policy urgency surrounding the opioid epidemic. Federal databases such as the Centers for Disease Control and Prevention's (CDC) overdose surveillance and SAMHSA's treatment facility directories provide high-resolution, publicly available spatial data, enabling fine-grained analyses rarely possible elsewhere. The national prioritization of the opioid crisis has also channeled research funding and policy attention toward this issue, creating a disproportionate body of US-based GIS evidence compared to other MH domains or regions.

## Gaps in evidence and future scope of use

While GIS offers powerful tools for studying MH service delivery, existing GIS research in MH is constrained by methodological

simplifications that limit cross-context transferability. Many studies assessing temporal accessibility measure travel time in terms of drive times to the nearest facility, implicitly assuming uniform transportation modes and potentially overlooking barriers faced by individuals reliant on public transport. The absence of measures that capture economic or cost-related barriers (such as transportation costs, time lost from work or out-of-pocket expenses) can lead to overestimation of true or effective access, as financial burdens may remain prohibitive despite apparent geographic proximity. Similarly, studies examining service availability often assume that proximity equates to access, ignoring capacity constraints, wait times or service saturation. It is also important to consider that the relevance of geographic location could differ across service types: For emergency services, such as opioid overdose treatment, rapid access is critical, whereas for non-emergency MH services, factors like privacy, stigma or patient comfort may make discrete or neutral service locations preferable to simply prioritizing proximity. Additionally, an exclusive focus on geographic distance may fail to capture other determinants of service use, such as stigma, privacy concerns or service acceptability, as highlighted by Cantor et al. (2022), who demonstrated that mapping services without considering payment acceptance could misrepresent true access.

Considering cultural or behavioral determinants of service delivery or integrating multiple dimensions (such as triangulating service utilization with availability) can provide a more accurate picture of true treatment capacity and better reflect the complexity of real-world service provision. Additionally, qualitative research can help elucidate the socio-cultural mechanisms underlying spatial patterns of service delivery, offering nuanced explanations for disparities observed through GIS analyses. Most GIS studies in the review offer static, cross-sectional snapshots of accessibility, overlooking how service reach and population mobility shift over time in response to policy changes, service expansion or closure and seasonal fluctuations in demand. Integrating longitudinal spatial analyses could help capture these temporal dynamics, offering a more realistic representation of equity in access to MH services.

More than 90% of the studies included in this review were conducted in high-income or upper-middle-income countries. The few studies conducted in LMICs leveraged existing administrative datasets or community surveys to generate actionable insights, demonstrating that creative use of available resources can support service planning and policy decisions. Future research in LMICs could build on these approaches by integrating multiple data sources, using open-source geographic data or applying community-driven mapping to expand GIS applications.

A major gap observed in this review was the lack of GIS usage in designing or monitoring trials related to MH service delivery. GIS can optimize recruitment strategies for clinical trials by identifying and targeting specific geographic areas with high prevalence of MH conditions or low service utilization. This can improve inclusivity of trial samples and reflect real-world dynamics (Krzyzanowski et al., 2019). Furthermore, GIS can help understand and address geographic barriers to participation and retention in trials, such as transportation difficulties or lack of local resources, increasing the external validity of trials (Arnold et al., 2024). It could also help in adopting more pragmatic approaches to trials, by informing adaptive trial designs and allowing for dynamic allocation of resources based on geographic disparities in service access (Savoca et al., 2017). It can be used to plan and monitor the delivery of community interventions within a trial (Nadkarni et al., 2024). For example, it can help ensure equitable distribution of resources across different geographic areas and track intervention implementation in real time.

There is also scope for expanding GIS usage in MH implementation research, specifically in leveraging it to support integration and coordination of MH services across different sectors (e.g., health care, social services and education). Mapping the distribution of services and identifying gaps in coverage can help improve service linkages and reduce fragmentation of care. It can also identify coordination gaps between primary care, specialized MH facilities and social support systems (Khashoggi and Murad 2020). GIS can enable monitoring of implementation outcomes of new programs and policies, providing real-time data on service delivery, utilization and outcomes. This information can be used to identify implementation challenges and make necessary adjustments to improve program effectiveness, helping us learn on the go and fundamentally transforming implementation research (Scotch et al., 2006; McGinty et al., 2024).

The visual impact of GIS mapping has a strong potential in shaping public health policies and advocacy strategies (Davenhall and Kinabrew, 2012; Manjunatha et al., 2024). Geospatial representations of treatment gaps, inequities and high-risk zones can provide compelling evidence to justify targeted funding allocations for MH infrastructure in underserved areas. GIS-based spatial equity audits could build a case to demand adjustments of service coverage to ensure marginalized communities are not disproportionately affected by service unavailability (Sharma and Ramesh, 2024). Participatory or community-driven mapping can also be used to advocate for policy reforms addressing systemic disparities and decentralization of MH services, ensuring that rural and remote populations have better access to care (Douglas et al., 2020).

Governments, funders and policymakers can take concrete steps to harness the potential of GIS for equitable MH service delivery. Integrating GIS into national health information systems could enable continuous monitoring of geographic inequities in MH service provision. Training policymakers and planners to interpret and apply GIS data in decision-making can help bridge the gap between technical analysis and governance. In parallel, funding agencies should invest in GIS-based implementation research (particularly in LMICs) promoting the use of open-source tools and participatory, community-engaged approaches.

Another consideration for future research and practice could be the use of interdisciplinary approaches in studying MH service delivery. The intersection of GIS with machine learning and MH sciences offers promising avenues for predictive analytics and precision MH (Kamel Boulos et al., 2019; Li and Ning, 2023; Fadiel et al., 2024). For example, spatial–temporal artificial intelligence (AI) models could predict future service demand based on socio-economic shifts, urbanization trends or climate change effects. Integration with mobile health (mHealth) tools could personalize treatment pathways based on an individual's geographic constraints. Legal and policy studies could utilize GIS to assess the impact of health policy changes on service accessibility over time.

## Limitations of the review

There are a number of limitations to our findings and review process. Our findings are presented descriptively as is typical for scoping reviews. We did not include gray literature or publications in languages other than English in our search, which may bias our results. Finally, we restricted the scope of the review to healthcare services, excluding preventive or promotional care delivered in other settings.

## Conclusion

GIS has the potential to emerge as a powerful tool in MH research, particularly in mapping disparities, informing service delivery and identifying high-risk zones. However, the existing literature remains concentrated in high-income settings, underscoring the need for context-specific applications in LMICs. Additionally, expanding GIS use in trial design, implementation science and policy advocacy could help bridge critical gaps in MH service delivery, ensuring more equitable and data-driven decision-making. We hope this scoping review provides researchers, policy-makers and service providers with an orientation to the current scope of GIS applications in MH service delivery and offers a foundation for advancing this work in diverse and underrepresented contexts.

**Open peer review.** To view the open peer review materials for this article, please visit http://doi.org/10.1017/gmh.2025.10088.

**Supplementary material.** The supplementary material for this article can be found at http://doi.org/10.1017/gmh.2025.10088.

**Data availability statement.** Data availability is not applicable to this article as no new data were created or analyzed in this study.

**Author contribution.** AN, VP, RV and DRS substantially contributed to the conception or design of the work. BB, RP, AKS, MGP, LF, YG and AF substantially contributed to the acquisition, analysis or interpretation of data for the work. BB, RP, AKS, MGP, LF, YG, DRS, AF, RV, VP, CL, CG, UB and AN drafted the work or revised it critically for important intellectual content. BB, RP, AKS, MGP, LF, YG, DRS, AF, RV, VP, CL, CG, UB and AN finally approved the version to be published. BB, RP, AKS, MGP, LF, YG, DRS, AF, RV, VP, CL, CG, UB and AN agreed to be accountable for all aspects of the work in ensuring that questions related to the accuracy or integrity of any part of the work are appropriately investigated and resolved.

**Financial support.** This study was a part of the IMPlementation of evidence-based facility and community interventions to reduce the treatment gap for depRESSion (IMPRESS) program that has been funded through a grant from the National Institute of Mental Health (NIMH), United States (Grant Number R01MH115504).

**Competing interests.** The authors declare none.

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
