## [Reviewer Report]

Overall, this review provides a comprehensive and timely overview of Geographic Information Systems (GIS) in mental health service delivery. This field offers significant potential for exploration due to its policy and practice implications. Although previous work exists, synthesis has been limited. Grounded in Joanna Briggs Institute methodology, the paper benefits from a pre-registered protocol (OSF DOI provided). The inclusion of 58 studies highlights significant trends, such as the overrepresentation of data from wealthier nations. It lays a solid foundation for further research. Professionals in global mental health, health systems, and digital health will find this manuscript useful. However, some areas would benefit from adjustments to optimise clarity, transparency, and impact.

Major Issues

Framing and Originality (Introduction)

The introduction does a decent job of placing GIS in the broader context of public health. However, it might benefit from a stronger emphasis on *why* its application in mental health service delivery has lagged behind other areas of healthcare. Clarify the added value. What does this review provide that Smith-East & Neff (2020) do not? Consider positioning it as a primary focus: addressing availability, accessibility, and utilisation globally, across the board.

Scope and Definitions (Methods)

Definitions of availability, accessibility, and utilisation are provided. However, they sometimes become unclear (for example, is *this* accessibility, or *that* utilisation?). Clarify the definitions and ensure consistent application throughout Results and Discussion.

Transparency of Screening and Selection

The PRISMA flow diagram is there, which is good. But, adding more detail about *why* full-text articles were excluded would improve reproducibility. A supplementary table listing excluded studies, *with reasons why*, would be helpful.

Depth of Critical Analysis (Results and Discussion)

The Results section is mainly descriptive, which is fine for a scoping review. The Discussion, however, could go deeper. For instance:

Why *specifically* are GIS applications so focused on opioid-related studies in the US?

What methodological weaknesses (e.g., drive-time measures, a lack of integrated cost/accessibility data) limit the extent to which we can apply these findings?

How do cultural and healthcare system differences impact the transfer of GIS methods from one location to another?

Equity Considerations

Equity keeps emerging as a theme, but we could further develop the analysis. How does GIS specifically highlight inequities in LMICs, and what implications does this have for implementation science and health policy in the future? Explain *how* GIS might reduce inequalities in mental health care access, especially in settings with limited resources.

Policy and Practice Recommendations

The conclusion makes some bold claims about GIS being a “powerful tool.” That’s fair, but the recommendations could be more concrete. Offer some actionable steps. For example, integrating GIS into national health information systems, training policymakers, or prioritising research funding in LMICs.

Minor Issues

Language and Readability

Generally, the language is strong. However, sentences can be lengthy with many references packed in. Editing for conciseness could enhance the flow.

Terminology Consistency

The manuscript jumps between “substance use disorders,” “opioid use disorder,” and “opioid-related overdose.” Consistently use one term throughout.

Figures and Tables

Table 1 is very detailed, possibly to an excessive degree. Consider splitting it into multiple tables (accessibility, availability, utilisation, etc.) to improve clarity.

References

Some recent digital health and GIS methodological references (2022–2024) should be added. Focus particularly on AI-enhanced spatial modelling and open-source GIS. tools that are appropriate for LMICs.

---

## [Reviewer Report]

This paper provides an overview of the use of GIS in analysing the availability, accessibility, and utilisation of mental health services. The scoping review appears to have been well-conducted, following recommended practice for scoping reviews and using a comprehensive search strategy. However, I believe that some edits to the manuscript could make it into a much more useful resource for the field.

The introduction introduces the concepts of availability, accessibility, and utilization. However the results section also includes uses multiple other related concepts, including service coverage/treatment coverage, help-seeking, various dimensions of equity, and treatment choices. It would have been helpful to more clearly spell out in the introduction how these fit into your taxonomy of service delivery dimensions, potentially using a figure or diagram to illustrate the theoretical framework used to organise the review and show which of the various terms used in the included studies are considered to be synonymous or overlapping.

The aims and methods are generally clear and well-described. The aims could be rephased slightly to avoid ambiguity (lines 147-149); i.e. specifying that you are examining the literature on the use of GIS as applied to mental health care (rather than exploring the use of GIS in analysing healthcare and the literature on mental health conditions). I was surprised by the inclusion of qualitative studies, since the phenomena of interest are quantitatively defined, so I think this needs some justification (and if there is genuinely a role for GIS in qualitative studies of mental health service availability/accessibility/utilization but no such studies were found then this deserves some mention in the discussion). I was curious about the two studies that were excluded because they were not focused on the three dimensions of service delivery – what was GIS used for in these cases? Spelling this out could shed light on the boundaries of how these terms are operationalised and whether relevant literature exists that uses alternative conceptual frameworks.

The individual results are clearly described, but could be organised in a clearer and more logical way. The accessibility heading is repeated twice. Some sub-headings could be rephrased to be more precise (e.g. “equity of services” is repeated under various sections – edit to be clear which aspect of equity this refers to in each case? Services can obviously be equitably distributed in space but still not be equitable in the service that they provide). Linked to the point above, the conceptual distinction between utilization, help-seeking, treatment coverage, and treatment choices is not always clear and could be made more explicit. Elaborating on the conceptual framework further and aligning this with the organisation of results and corresponding sub-headings would help the reader to connect the content and follow the flow of the results. In some cases it’s unclear to me why a study has been categorised under the heading used; e.g. should the Holmes et al (2022) paper have been included under the “impact” heading, since it investigated opioid overdose survival? Some of the studies classified under “impact on treatment choices” seem to investigate accessibility of different types of facility rather than patient choices about type of treatment, based on the description in the text (e.g. the Alibrahim et al 2022 study, lines 444-447; the Kleinman 2020 study, lines 453-457; and possibly the Charlesworth 2024 study; lines 457-460). It was also unclear how “cost of travel” was operationalised spatially (Han and Stone 2007) – presumably most studies that focussed on travel cost were excluded as they did not use geographic methods (in which case I would not highlight this as a finding).

There are many missed opportunities in the discussion to highlight what GIS can and can’t do to advance the field of mental health services research, so I think that rewriting this would make the paper substantially more useful to other researchers considering applying GIS to study access to mental health services. It would be useful for the first paragraph of the discussion to briefly list the underexplored areas identified (rather than simply state that there were identified). Similarly the statement that the review has identified emerging methodological directions that can advance mental health service research would be more substantive if these were elaborated; perhaps this could be rephrased to say that in the following paragraphs you will discuss the potential applications of the methods used, to show which have promise in advancing mental health service research beyond the limited conditions and geographic regions studied to date. It would be very helpful for the discussion to briefly spell out what the methods mentioned in the results can do; for example, what sorts of research questions require spatial regression techniques to answer? What questions can E2SFCA methods address, when applied to mental health services research?

At the moment there is no discussion of the assumptions made in the studies and the potential for GIS studies to mislead if these assumptions are not critically examined (e.g. do some methods assume that everyone drives, and would they come to different conclusions if they were based on assessing public transport routes? Are the datasets used to measure utilization etc generally reliable? Do the studies assume that services have unlimited capacity if they are geographically accessible (i.e. having a service within a given radius is taken to indicate availability regardless of how over-subscribed they may be) and if not how did they factor service capacity in to their models? To what extent to the included studies capture both public and private services, and use of general health services for mental health reasons (e.g. GP consultations about mental health) and what should future research learn from the methods used in the literature to date to conduct robust analyses of access to mental health care? There is always a risk that enthusiasm for shiny new methods distract from more basic questions, leading to misleading findings when key assumptions are overlooked because of the sophistication of the analysis techniques applied. This review could help to improve GIS research on mental health services by noting not only the potential applications of these methods but also the limitations of the methods used, and provide a reminder that looking at distance alone can produce misleading results (as demonstrated by the Cantor 2022 study that explored whether the available facilities accepted the forms of payment that service users had access to).

I would have liked to see some recommendations for how to expand the use of GIS in LMIC, potentially by exploring how the included studies managed to employ it despite limited existing databases, to help others to apply similar approaches or build on these methods. It would also be helpful to discuss the ways in which geographic location matters in different ways for different conditions, especially given the predominance of opoid overdose treatment studies. Treating overdoses relies on people being able to reach a service quickly in an emergency situation, whereas the way in which people interact with non-emergency mental health services differs substantially. In the latter case, having services located in a discrete location where they’re not likely to bump into people they know might be preferable to having services as close as possible to patients’ homes.

The point about GIS use in trials is a good one, but could perhaps have been highlighted earlier (by including study types in the table of study characteristics and noting the absence of trials in the text on study characteristics) so that this finding isn’t introduced for the first time in the discussion. The statement that about one third of studies explored objectives related to resource management and planning is confusing; how was this classified? I’m not sure I could identify which these are based on the manuscript. Surely the ultimate point of all of the analyses is to inform resource allocation decisions? Similarly, I’m not sure how studies were classified as using GIS for program evaluation (documenting the impact of the Medicaid model of payment is arguably not program evaluation). Note that the finding that >90% of studies were from high-income or upper-middle-income countries is not a limitation of the review; it is a finding based on the existing evidence (and the idea that generalisability of findings is limited by this doesn’t make sense in the context of a scoping review, since you are not drawing conclusions about a specific question but simply mapping the evidence base).

Finally, there are a few issues of grammar and formatting that need addressing prior to publication:

• The numbering of sub-headings within the results section is not consistent (I. Accessibility, II. Availability, III. Accessibility, no number for the “Utilization” sub-heading)

• Some minor typos need correcting (e.g. line 162 should say “this encompasses”). The sentence on line 490 needs to be rephrased (“by correlating spatial healthcare access with socioeconomic indicators” - replace with “by assessing associations between”?)

• References are needed for the software used (Endnote/Covidence)

• I would recommend including a brief definition of any terms that will not be familiar to those who are new to spatial methods (e.g. Euclidean)

• Date missing for Sharma and Ramesh reference on line 546

---

## [Editor Report]

We have now received comments from reviewers on your manuscript. Based on their comments and suggestions we have reached a decision and recommend major revisions to your manuscript. Attached are reviewer comments for you to address and respond to.

---

## [Reviewer Report]

Thank you for the thoughtful responses to my comments. I believe that the edits have addressed all of my feedback and am happy to recommend the manuscript for publication.

---

## [Editor Report]

Dear Authors 

We have reviewed the revisions you made to the manuscript and we accept the revised manuscript for publication.

Regards

Siham